# Taxonomy Completion via Implicit Concept Insertion

## ABSTRACT

High quality taxonomies play a critical role in various domains such as e-commerce, web search and ontology engineering. While there has been extensive work on expanding taxonomies from externally mined data, there has been less attention paid to enriching taxonomies by exploiting existing concepts and structure within the taxonomy. In this work, we show the usefulness of this kind of enrichment, and explore its viability with a new taxonomy completion system ICON (**I**mplicit **CON**cept Insertion). ICON generates new concepts by identifying implicit concepts based the existing concept structure, generating names for such concepts and inserting them in appropriate positions within the taxonomy. ICON integrates techniques from entity retrieval, text summary, and subsumption prediction; this modular architecture offers high flexibility while achieving state-of-the-art performance. We have evaluated ICON on two e-commerce taxonomies, and the results show that it offers significant advantages over strong baselines including recent taxonomy completion models and the large language model, ChatGPT.

## CCS CONCEPTS

• **Computing methodologies** → **Ontology engineering**; **Semantic networks**.

## KEYWORDS

Taxonomy Completion, Taxonomy Enrichment, Ontology Engineering, Text Summarisation, Pre-trained Language Model

**ACM Reference Format:**
Anonymous Author(s). 2018. Taxonomy Completion via Implicit Concept Insertion. In *Proceedings of Make sure to enter the correct conference title from your rights confirmation emai (Conference acronym 'XX)*. ACM, New York, NY, USA, 11 pages. https://doi.org/XXXXXXX.XXXXXXX

## 1 INTRODUCTION

A taxonomy is a hierarchical knowledge graph where edges in the graph represent *is-a* relationships between concepts. It takes the form of a directed acyclic graph (DAG), and is sometimes simplified to a tree. For a wide range of domains such as natural sciences, medicine, web search, and e-commerce, taxonomies form the backbone of domain knowledge, thus serving many downstream applications such as query answering, natural language understanding, recommendation, and information retrieval.

In order to meet the needs of these applications, taxonomies are expected to be complete and accurate. Completeness refers to

the nodes in a taxonomy (often called concepts) covering as many concepts relevant to the underlying domain as possible, and accuracy refers to the edges in a taxonomy (often called subsumption relations) correctly capturing the *is-a* relationships in the domain, possibly through transitive closure. There is a large body of research oriented at completing taxonomies (see related works in Section 2), but most of these studies focus on deriving new concepts from external resources. However, taxonomies can also be enriched from information within themselves. This is often observed as a concept whose existence is implied by the structure of the taxonomy, but is currently missing from it [10]. We call these concepts **implicit concepts**. Consider a typical segment of an e-commerce taxonomy with 7 concepts, as shown in Figure 1. The top concept, "Clothing, Shoes, & Accessories (CSA)", is divided along gender at level 2, and both subconcepts are further divided along product type at level 3. The concepts at level 3 imply the existence of standalone "clothing" and "shoes" concepts, as illustrated in Figure 1, but these concepts are in fact missing.[1] What is *not* a remedy to this problem is to replace the level 2 concepts with the two inferred concepts, as the current Men's and Women's CSA concepts will then be lost. Instead, one must break the tree structure to add these new concepts. This is clear when we consider more profound examples such as the one illustrated in Figure 2, where the implicit concept reveals semantic connections between concepts in distant branches of the taxonomy. In this case the implicit concept "Sporting Shoes" reveals connections between sub-concepts of CSA and Sporting Goods; it is a sub-concept of both these level-1 concepts, and its sub-concepts form part of the intersection between these level-1 concepts (we show only a fraction of these sub-concepts due to space limits). Similar to these examples, most implicit concepts are intermediate nodes that reflect alternative ways to organise the hierarchy. However, the pool for possible intermediate nodes is exponential in terms of taxonomy size. Therefore, careful identification of intermediate nodes is required in order to find useful implicit concepts that can improve the taxonomy's quality and benefit downstream applications.

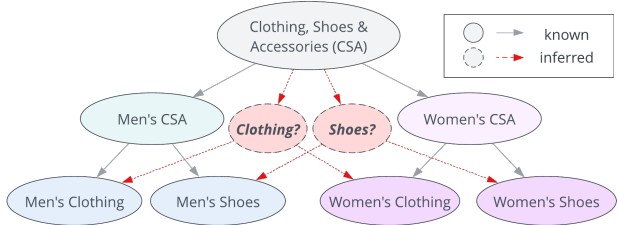

**Figure 1: Implicit concept examples ("Clothing" and "Shoes") in an e-commerce taxonomy**

In this work we aim to address the issue of finding such implicit concepts and inserting them in the taxonomy; we call this problem

---

[1] Browse this segment at https://ebay.com/b/11450

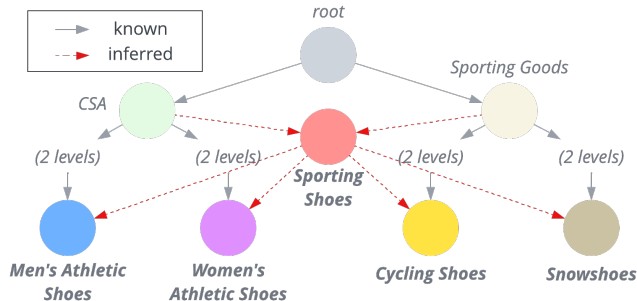

**Figure 2: Implicit concept "Sporting Shoes" that connects concepts spreading across different level 1 branches**

*implicit taxonomy completion* (see Section 3.2 for the formal definition). To the best of our knowledge, there is no existing work that directly tackles implicit taxonomy completion. GenTaxo [38] is closely related in that it generates new concept names and predicts whether the new concepts can be inserted into given positions in the taxonomy. However, it doesn't identify where a new concept might be useful or where to insert it in the taxonomy; this information must be provided to the system in the form of a complete set of parent and child concepts for the proposed new concept.

Implicit taxonomy completion can be decomposed into three sub-tasks: identifying potentially useful implicit concepts, naming implicit concepts, and inserting them in the taxonomy. We propose a novel taxonomy completion system, **I**mplicit **CON**cept Insertion (ICON), that integrates solutions for each of the three sub-tasks. First, we use ideas from *entity retrieval* to address the identification sub-task; specifically, we use a KNN algorithm with BERT embeddings to identify clusters of existing concepts that might correspond to an implicit concept. Second, we use ideas from *text summarization* to address the naming sub-task; specifically, we use T5 [27] to create a label for the union of the concepts selected in the first sub-task. Finally, we use ideas from *taxonomy completion* to address the insertion sub-task; specifically, we use a modified enhanced traversal algorithm that employs BERTSubs [4] to perform subsumption tests. Note that the traversal might identify an existing equivalent concept, in which case ICON simply adds any missing *is-a* links.

We have evaluated ICON on two large real-world taxonomies, eBay and the concept hierarchy extracted from AliOpenKG. Our experiments indicate a dramatic improvement over existing techniques and baselines. We also examine in ablation studies the effects of varying hyperparameters and search options.

Our contributions are summarised as follows:

1. Proposal of the implicit taxonomy completion task based on the observation of real-world taxonomies.
2. Design of a flexible framework for implicit taxonomy completion.
3. Implementation of the framework in the ICON system with novel optimisation and language model fine-tuning methods for higher scalability.
4. Extensive evaluation demonstrating ICON's strength at implicit taxonomy completion.

## 2 RELATED WORK

### 2.1 Taxonomy Construction

Taxonomy construction is the building of a taxonomy, often from scratch, using a corpus. Traditionally, taxonomy construction can be divided into concept taxonomy construction, which applies semantic inductions on candidate concepts to gradually build hierarchies, and topic taxonomy construction, which features some clustering on keywords [33]. Early taxonomy construction methods are usually graph-based [14, 23, 26], or distribution-based [36]. These methods use either graph models or probabilistic models to propagate taxonomic knowledge, and the atomic semantic induction is usually based on simple lexicosyntactic features. Later methods incorporate word embeddings for better semantic understanding [15, 29]. Reinforcement learning has also been applied for taxonomy construction, where the reward is set to a similarity metric between the constructed taxonomy and a gold-standard taxonomy [20].

### 2.2 Taxonomy Expansion

Taxonomy expansion inserts new concepts into existing taxonomies. Large open source taxonomies such as WordNet [21] and MeSH [16] have led to increased attention on taxonomy expansion, since improving upon a well curated taxonomy is easier than building one from the ground up. In contrast to taxonomy construction, taxonomy expansion emphasises the interaction between new concepts and existing concepts. An early method known as APOLLO [30] uses graph knowledge propagation algorithms to predict category memberships for text mentions. Most later methods are embedding-based, with the central idea of first encoding both the new concept and the known concepts with embeddings, then decoding to obtain a concept prediction, as practiced in ETF [32]. Further improvements include Arborist [19] which consideres implicit edge semantics, STEAM [37] which builds the mini-path corpus to improve encoding quality. Other recent models explore different ways to utilise the tree structure or multimodal information as features to improve ranking. For instance, TaxoExpan [28] injects neighbourhood information of the anchors, forming a mini-graph known as Egonet; HyperExpan [18] is an improvement that shares a similar philosophy but uses hyperbolic embeddings which is better at encoding hierarchical structures [5, 24]; HEF [34] builds the ego-tree which consists of all the known ancestors and children of a node, in an effort to maximise hierarchical information visible to the model. TaxoExpan and HyperExpan use Graph Neural Networks for decoding, while HEF uses pre-trained language models (PLMs).

### 2.3 Taxonomy Completion

The main drawback of taxonomy expansion is that the task only considers superconcepts of the candidate. It does not express the full ground truth when the candidate is not leaf. Therefore, TMN [39] spearheads a more realistic task known as taxonomy completion, which considers both superconcepts and subconcepts of the candidate. The primary-auxiliary scorer structure of TMN is used by many later models. TaxoEnrich [12] adopts the TMN structure,

injects sibling information, and replaces the vector dot product metric with PLMs. QEN [35] adds even more information and evaluates (candidate, parent, child, sibling) quadruplets instead of triplets.

Together with the taxonomy expansion methods, these methods share a common pattern in utilising the contextual data: feed a large set of features into the model, obtain a representation and decode this representation for a score. However, both the model and the corpus quickly grow in size, increasing the computational overhead that is required to process marginally relevant features. Therefore, we try to handle contextual data differently in our work: we use minimal information for a single prediction, and perform multiple predictions to *search* for the optimal subconcepts and superconcepts, where the search is naturally guided by taxonomic hierarchy.

### 2.4 Implicit Concepts

Despite the common presence of implicit concepts, very little research has addressed this issue. Existing work has focussed on determining whether a *mention* (in some external source) corresponds to a concept missing from a knowledge base [6, 7, 11], but this approach can only spot implicit concepts one at a time and only when a candidate is provided from an external source.

## 3 PRELIMINARIES

### 3.1 Taxonomy

A *taxonomy* $\mathcal{T} = (\mathcal{N}, \mathcal{E})$ is a transitively reduced DAG where the nodes $n \in \mathcal{N}$ are concepts, and the edges $e \in \mathcal{E}$ are *is-a* relations. We say $n_1$ is a *parent* of $n_2$, or equivalently $n_2$ is a *child* of $n_1$, if $(n_2, n_1) \in \mathcal{E}$. The collection of all parents and children of a concept are denoted $p(n)$ and $c(n)$ respectively. The *ancestors* of $n$, denoted $A(n)$, is the set of all nodes reachable from $n$ in the DAG. Conversely the *descendants* of $n$, $D(n)$, is the set of all nodes reachable from $n$ when all the edges in $\mathcal{E}$ are reversed. $n_1 \in (A(n_2) \cup \{n_2\})$ is also denoted as $n_2 \sqsubseteq n_1$, which reads "$n_1$ *subsumes* $n_2$". In particular, this subsumption is *direct* if $n_1 \in p(n_2)$. Each concept should have a *label*, consisting of the concept's name (a text string) and optionally some additional natural language description. Where there is no ambiguity we will use $n \in \mathcal{N}$ to refer to both a concept and its label. The *top* concepts $\top_{\mathcal{T}}$ and *bottom* concepts $\bot_{\mathcal{T}}$ of $\mathcal{T}$ are defined as the concepts without parents and children respectively. An *intermediate concept* in $\mathcal{T} = (\mathcal{N}, \mathcal{E})$ is defined as a non-singleton subset of $\mathcal{N}$. We define a taxonomy $\mathcal{S} = (\mathcal{N}_{\mathcal{S}}, \mathcal{E}_{\mathcal{S}})$ to be a *sub-taxonomy* or *subgraph* of a taxonomy $\mathcal{T} = (\mathcal{N}_{\mathcal{T}}, \mathcal{E}_{\mathcal{T}})$ if $\mathcal{N}_{\mathcal{S}} \subseteq \mathcal{N}_{\mathcal{T}}$ and $\mathcal{E}_{\mathcal{S}} \subseteq \mathcal{E}_{\mathcal{T}}$.

### 3.2 Problem Statement

Given a taxonomy $\mathcal{T} = (\mathcal{N}, \mathcal{E})$ and a candidate concept label $q$ (candidate for shorthand), *taxonomy completion* is the task of inserting $q$ into $\mathcal{T}$. It consists of finding $q$'s parents $p(q)$ and children $c(q)$ from $\mathcal{N}$, and replacing $\mathcal{T}$ with the transitive reduction of $\mathcal{T}' = (\mathcal{N} \cup \{q\}, \mathcal{E} \cup \{(q, p) | p \in p(q)\} \cup \{(c, q) | c \in c(q)\})$, such that $\mathcal{T}'$ is a DAG. Notice that the inserted concept $q$ could coincide with an existing concept in $\mathcal{N}$, in which case we add extra edges to $\mathcal{T}$ but do not add extra nodes.

Implicit taxonomy completion is a special case of taxonomy completion where the candidates are generated from the existing taxonomy. An implicit concept within our scope of interest is simply an intermediate concept $I \subseteq \mathcal{N}$. We define implicit concept insertion as follows: given a taxonomy $\mathcal{T} = (\mathcal{N}, \mathcal{E})$ and an intermediate concept $I \subseteq \mathcal{N}$, generate a label $q$ for $I$'s most specific super-concept and insert $q$ into $\mathcal{T}$. Implicit taxonomy completion consists of (repeatedly) identifying "relevant" intermediate concepts and inserting them via implicit concept insertion. An intermediate concept is relevant if the resulting implicit concept insertion usefully enriches the structure of the taxonomy; this must be empirically determined and may be application dependant.

## 4 METHODOLOGY

We will start by outlining the overall architecture of ICON, which includes its three main components, stepwise workflow, relations between its outer loop and inner loop, and three modes of operation. Next, we present the self-supervised training scheme for each sub-model including its data and training objectives. These models can be freely updated or replaced for later improvements.

### 4.1 Overall Architecture

ICON adopts an iterative workflow that consists of two nested loops. The outer loop ranges over the taxonomy and creates clusters from which intermediate concepts are built. The inner loop ranges over subsets of each cluster, generates intermediate concepts and performs implicit concept insertion on them. Figure 3 illustrates the basic workflow of ICON, and Figure 4 provides a running example of how ICON processes a single seed concept.

*4.1.1 The outer loop.* The outer loop (steps 1 and 2 in Figure 3) covers the entity retrieval part of the task. It starts with a seed $s$ that is either given as input or randomly selected. $s$ can be either a concept in the taxonomy or a textual phrase, and all operations in this iteration happen in the semantic vicinity of $s$. A sub-model, which we call the KNN model because our implementation of it uses k-nearest neighbours on embeddings, retrieves the $k$ concepts most similar to $s$, denoted as its cluster $C(s)$. A practical setting of $k$ is between 5–10. The reason we select similar concepts is to increase the chance a subset built from these concepts identifies an implicit concept. In Figure 4a, we illustrate as an example the case where $s$ = Men's Vintage T-Shirts. Setting $k = 5$, the KNN model retrieves the following relevant concepts (shown as light green nodes): Men's T-Shirts, Men's Equestrian Shirts, Men's Western Show Shirts, Vintage Sports Shirts. These four concepts together with $s$ form the cluster.

*4.1.2 The inner loop.* The inner loop (steps 3 to 6 in Figure 3) receives $C(s)$ and covers the other two sub-tasks: label generation and concept insertion. Given a cluster $C(s)$, it first enumerates all the subsets of $C(s)$ up to size $m$. There is the option to limit the subsets to those that include $s$, which leads to $O(|k^{m-1}|)$ subsets as opposed to $O(|k^m|)$ when we do not impose that limit. For tractable computation $m$ is usually set to 2 or 3. For each subset $\{c_1, c_2, \ldots, c_m\}$, the GEN model is called to generate a new label $q$ that summarises $\{c_1, c_2, \ldots, c_m\}$. In our example, we set $m = 2$ and one of the subsets thus built is {Men's Vintage T-Shirts, Men's Western Show Shirts}. Our GEN model then produces "Shirt" as its label, indicated by the blue arrows in Figure 4a. The new labelled

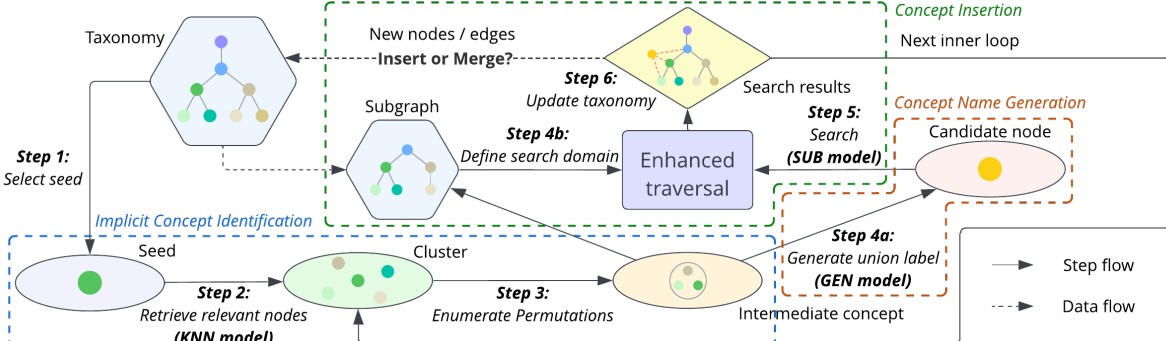

**Figure 3: Workflow of ICON. In steps 1 - 2, it retrieves a cluster of nodes for a given seed. In steps 3 - 6, it traverses some subsets of the cluster, generates a labelled concept for each subset and inserts it into the taxonomy. After looping steps 3 - 6 over all the intended subsets, it goes back to step 1 and selects a new seed. Steps 1 - 2 form the outer loop, and steps 3 - 6 form the inner loop.**

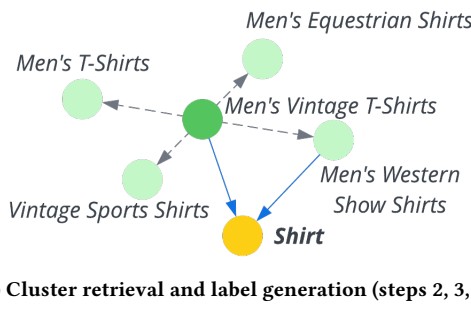

**(a) Cluster retrieval and label generation (steps 2, 3, 4a)**

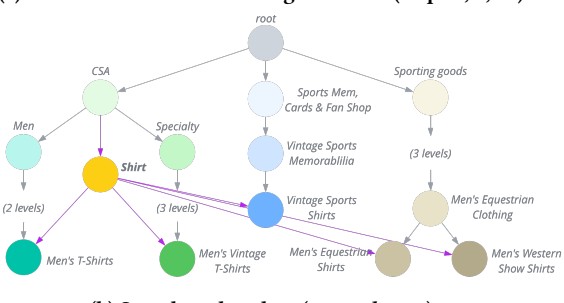

**(b) Search and update (steps 4b, 5, 6)**

**Figure 4: Example for the seed "Men's Vintage T-Shirts"**

*candidate* concept will then be inserted into the taxonomy using enhanced traversal.

*Enhanced traversal.* Enhanced traversal [1, 2, 9] can be understood as a two-stage Breadth First Search (BFS) that locates where a candidate concept $q$ should be inserted in the taxonomy. The first stage is top-down, searching for the lowest / most specific parents of $q$. The second stage is bottom-up and searches for the highest / most general children of $q$. Both searches use the hierarchy to prune branches; e.g., if a node has already been determined to not subsume $q$, then neither can any of its children subsume $q$. If the two searches intersect at a concept $D$, then $D$ is both a parent and a child of $q$ and we have $q \equiv D$. Algorithm 1 gives pseudocode for the basic variant of enhanced traversal. In practice, we make several

modifications to the search algorithm to tackle the uncertainty of the neural-based SUB model and improve search speed:

- *Tolerance.* The standard enhanced traversal uses logical reasoning for the atomic subsumption test, which provides a semantically deterministic subsumption test. Here we use the PLM-based SUB model as a substitute. Using a neural model could exploit some likely subsumptions but also introduce inevitable errors. To compensate for errors we introduce the option to add tolerance to pruning, which allows the algorithm to explore up to depth $\tau$ below / above a node that has failed the subsumption test. High tolerance runs the risk of making search slow as more candidates are pushed into the queue to be visited.
- *Forceful inclusion of results.* Since $q$ is intended to represent a union of several concepts, it makes sense to automatically include these *base* concepts into $c(q)$. In addition, it makes sense to include the lowest common ancestor (LCA) of base concepts in the original taxonomy into $p(q)$. This is because the LCA must be at least as general as the union, assuming that GEN model provides a faithful representation. Turning this option off allows the SUB model's decisions to override the GEN model's assumptions.
- *Search space constraints.* A significant optimisation on the search is to limit the search space to a highly relevant subgraph of the whole taxonomy. Since we already have a good estimate of what could be relevant to $q$ (the base concepts, or the cluster from which the base concepts are selected), a natural choice of this subgraph is the subgraph *spanned* by the bases (cluster) *up* to the LCA. More formally, this is the subgraph induced by all nodes that transitively subsume at least one base (cluster member), and are transitively subsumed by the LCA of the bases (cluster). Restricting the search space has another safety benefit when forceful inclusion is enabled, since the SUB model might erroneously predict a descendant of a base concept to be in $p(q)$, or an ancestor of the LCA in $c(q)$, causing cyclic subsumption.

*Update taxonomy.* We consider three cases when processing the search results $(p(q), c(q))$:

---

**Algorithm 1:** Enhanced traversal

**Data:** Taxonomy $\mathcal{T}$, candidate node $q$

**Result:** Parents $\mathrm{p}(q)$, children $\mathrm{c}(q)$

1  $\mathrm{p}(q) \leftarrow \emptyset$                     /* Top-down search */

2  visited $\leftarrow \emptyset$

3  queue $\leftarrow \top_{\mathcal{T}}$

4  **while** queue $\neq \emptyset$ **do**

5      $x \leftarrow \mathrm{pop}(\text{queue})$

6      **if** $x \notin$ visited **then**

7         visited $\leftarrow$ visited $\cup \{x\}$

8         **if** SUB_MODEL$(q, x) = $ true **then**

9            $\mathrm{p}(q) \leftarrow \mathrm{p}(q) \cup \{x\} \setminus \mathrm{A}(x)$   /* Only keep the most specific parents */

10           **for** $c \in \mathrm{c}(x)$ **do**

11              push(queue, $c$)

12 $\mathrm{c}(q) \leftarrow \emptyset$                  /* Bottom-up search */

13 visited $\leftarrow \bigcup_{p \in \mathrm{p}(q)} \mathrm{A}(p)$       /* Avoid graph cycles */

14 queue $\leftarrow \bot_{\mathcal{T}}$

15 **while** queue $\neq \emptyset$ **do**

16     $x \leftarrow \mathrm{pop}(\text{queue})$

17     **if** $x \notin$ visited **then**

18        visited $\leftarrow$ visited $\cup \{x\}$

19        **if** SUB_MODEL$(x, q) = $ true **then**

20           $\mathrm{c}(q) \leftarrow \mathrm{c}(q) \cup \{x\} \setminus \mathrm{D}(x)$   /* Only keep the most general children */

21           **for** $p \in \mathrm{p}(x)$ **do**

22              push(queue, $p$)

---

- *Reject.* When $\mathrm{p}(q) = \emptyset$, we say that $q$ has nowhere to go in the taxonomy and is therefore rejected. The system would then process the next intermediate concept generated in step 3 of Figure 3.
- *Insert.* When $\mathrm{p}(q) \neq \emptyset$ and $\mathrm{p}(q) \cap \mathrm{c}(q) = \emptyset$, we do not find any existing node that equals $q$. Therefore, we declare $q$ to be a new concept, and insert this concept with the predicted edges accordingly into the taxonomy.
- *Merge.* When $\mathrm{p}(q) \cap \mathrm{c}(q) \neq \emptyset$, we find that at least one existing node equals $q$. However, if we declare equality across all nodes in $\mathrm{p}(q) \cap \mathrm{c}(q)$, we will be effectively gluing several existing nodes together, which is usually wrong and can cause logical contradictions. Therefore, we keep only the one node in $\mathrm{p}(q) \cap \mathrm{c}(q)$ with the highest confidence score, denoted as $e$, merge it with $q$, and set other nodes to either a parent or a child of $e$, depending on which prediction has a higher confidence[2]. Notice that, in this case, we effectively perform *missing link prediction* on the taxonomy.

For our running example, we adopt a search space spanned by the cluster, obtaining a subgraph of 21 concepts. Our search on this subgraph returns the results illustrated in Figure 4b. The new concept Shirt will be inserted to the taxonomy under CSA (Clothing, Shoes and Accessories), with the five concepts in the

---

[2]Our neural SUB model provides a confidence score $s \in [0, 1]$ for each predicted parent or child. The confidence score for equivalence can be obtained by combining the individual scores for parent and child: $s_e = s_p s_c$.

---

cluster being its children. Its predicted subsumptions are marked with violet arrows. Notice that the five predicted child concepts are scattered across different branches in the taxonomy, but with our approach we can establish links across these branches via suitable intermediate concepts.

*4.1.3 Modes of operation.* ICON can function in three modes, depending on the task. *Auto* mode is fully automated and exploits the full workflow in Figure 3. The system keeps selecting new seeds from nodes that haven't been involved in any clusters yet, until having all of the original nodes of the taxonomy involved in at least one cluster. *Semiauto* mode is frequently used in evaluation settings, in which seeds are given by manual inputs, replacing step 1 in Figure 3. *Manual* mode is used for conventional taxonomy completion where candidate nodes are directly given by manual input, and only steps 4b and 5 (optionally 6) will be called.

## 4.2 Required Models

As mentioned above, the system depends on three models: KNN, GEN, and SUB. Their basic properties are summarised in Table 1. We now describe our implementation for each model.

**Table 1: Basic properties of the adopted models in ICON**

| Model | Input | Output | Function |
|---|---|---|---|
| KNN | $\mathcal{T}$, query concept or phrase | set of concepts | retrieve concepts similar to query |
| GEN | set of concepts | concept label | generate union label |
| SUB | $(n_1, n_2)$ | confidence score | predict whether $n_1 \sqsubseteq n_2$ |

*4.2.1 KNN model.* We use the cosine metric on contrastive learning and BERT-based embeddings to compute similarity scores. The model is trained with the supervised SimCSE [8] framework. For a taxonomy, we collect nodes that have more than one child to obtain all sibling instances. With careful arrangement, we organise the sibling pairs into mini-batches $(x_i, x_i^+)$, $i = 1, \ldots, N$ such that no $x_i$ and $x_j^+$ are siblings if $i \neq j$. The training objective is to minimise the following contrastive loss, summed over all mini-batches:

$$\mathcal{L} = -\sum_{i=1}^{N} \log \frac{e^{\mathrm{sim}(\mathbf{h}_i, \mathbf{h}_i^+)/\tau}}{\sum_{j=1}^{N} e^{\mathrm{sim}(\mathbf{h}_i, \mathbf{h}_j^+)/\tau}} \tag{1}$$

where $h_i$ denotes the hidden representation of $x_i$, $\tau$ is a temperature hyperparameter and $\mathrm{sim}(\mathbf{x}, \mathbf{y}) = \frac{\mathbf{x} \cdot \mathbf{y}}{\|\mathbf{x}\| \|\mathbf{y}\|}$ is the cosine similarity.

*4.2.2 GEN model.* T5 [27] is a transformer-based sequence to sequence language model. We leverage the capability of T5 to summarise concepts into an upper level of abstraction. Our training data follows the format of $\{\texttt{text:} n_1, n_2, \ldots, n_m, \texttt{ summary:} n_{LCA}\}$, where $n_1, n_2, \ldots, n_m$ are $m$ comma delimited node labels with $m$ ranging from 2 to 5, and $n_{LCA}$ is the label of the LCA of the nodes. Following standard T5 prompt templates, we prepend the task

prompt "summarize:" to all inputs (e.g., "summarize: Men's clothing, Women's clothing, Kid's clothing").

For 70% of the data, the nodes to be summarised are different children or grandchildren of the same LCA. However, for the other 30% we randomly corrupt one or several of the input nodes, so that the overall LCA becomes the global root node of the taxonomy. In these corrputed rows the reference summary is a special placeholder token which indicates that the input node combination is of poor quality. When ICON receives the placeholder token from step 4a in Figure 3, it rejects the intermediate concept immediately and proceeds to the next inner loop. From this type of training, the model learns the ability to discern implicit concepts from combinations of loosely relevant concepts (e.g., {smartphone models, smartphone cases, retail display cases}). The training objective is to minimise the cross entropy loss for language modelling [13] between the prediction and the reference.

*4.2.3 SUB model.* Subsumption prediction can be formalised as an instance of binary classification (whether the child-parent pair holds between the two concepts). We fine-tune BERT for training and inference. Child-parent pairs $(n_c, n_p)$ are used as positive samples, while negative samples are obtained by replacing the $n_p$ with a node that does not subsume $n_c$. Such a node comes from two sources: random sampling, which creates easily discernible negatives; and graph random walk from the true subsumer, which creates harder negatives. The model is fine-tuned on this dataset with a binary cross entropy loss. The SUB model is the most frequently called of all three sub-models, thus it demands a lightweight build in order to ensure inference speed. We adopt the BERTSubs model with its "isolated class" setting [4] which features fast inference while maintaining satisfying accuracy.

## 5 EVALUATION

Following our formulation and decomposition of implicit taxonomy completion, we design our evaluation based on the following research questions:

- **RQ1.** How well do the methods identify the implicit concepts in a given taxonomy?
- **RQ2.** Given an implicit concept, what are the qualities of the labels generated by the methods?
- **RQ3.** Given a concept label, can the methods accurately predict its relevant subsumptions for insertion?

These research questions correspond to the sub-tasks of implicit concept identification, concept name generation, and concept insertion, respectively.

As mentioned in Section 2, we are not aware of any existing system that can perform implicit taxonomy completion. The GenTaxo system can perform some of the relevant tasks, and we extended it to GenTaxo++ by integrating conventional taxonomy completion models as suggested in [38]. We tried three such models: TaxoExpan [28], TMN [39], and QEN [35]. We also used a general purpose large language model, ChatGPT[3], a GPT-like model [3] further trained with instruction tuning [25]. We use ChatGPT with appropriate prompts for each of the three sub-tasks.

---

[3]https://openai.com/blog/chatgpt

A fundamental assumption of taxonomy completion is that the given taxonomy does not represent the full truth (i.e., it is incomplete). Measuring precision is therefore very challenging because many concepts and subsumptions that are absent in the reference taxonomy could still be factually valid [31]. To address this problem, we use **human labelling** in the relevant experiments.

### 5.1 Experiment Setup

*5.1.1 Datasets.* We conduct the experiments on two large real-world taxonomies: the concept hierarchy extracted from the e-commerce ontology AliOpenKG[4], and eBay's product taxonomy[5]. Table 2 lists their basic properties. Since both taxonomies are trees, they possess one fewer edges than nodes.

**Table 2: Metadata of the taxonomies used in evaluation**

| Taxonomy | #Nodes | Max depth | Avg. depth | Language |
|----------|--------|-----------|------------|----------|
| eBay | 20,322 | 6 | 4.235 | English |
| AliOpenKG | 7,100 | 4 | 3.896 | Chinese |

For evaluation, we artifically create implicit concepts by "masking" some of the existing taxonomy concepts. For each taxonomy, we mark its *second-to-bottom* level nodes as either non-testing or testing in a 4:1 ratio. Nodes marked for testing will be removed from the taxonomy and their children will be directly connected to their grandparents. The original ontology can then be used as a gold standard for implicit concept insertion.

Both ICON and GenTaxo++ require training. While ChatGPT cannot be trained, its performance can be improved with examples of the task as demonstrations in the prompt [22]. Therefore, we split the datasets into training and validation branches. The validation branch is generated as follows: start from a random leaf node and traverse the taxonomy randomly until 10% of the nodes have been visited, then use the subgraph induced by the visited nodes as the validation branch. The training branch is then constructed from the remaining nodes by adding any edges needed to reconnect the taxonomy after removing the validation branch. We use the training branch to fine-tune each of the methods, or in the case of ChatGPT to provide it with examples.

*5.1.2 Task.* Our experiment consists of two stages. First, we attempt to recover the masked nodes and generate their labels. This corresponds to RQ1 and RQ2. For ICON, we use the semiauto mode where we input one seed for each masked node, selected randomly from the children of the masked node; for GenTaxo++, we provide *candidate positions* for each masked node, which are the sets of parent and child concepts derived from the masked nodes' original positions; for ChatGPT, we give one input for each masked node consisting of all its children, and prompt the model to either create intermediate concepts based on the input, or output a negative response if it cannot find any intermediate concept. We will provide the details of our prompts in Appendix C. Note that this gives a significant advantage to GenTaxo++ and ChatGPT as we provide

---

[4]https://kg.alibaba.com/overview/index.html
[5]https://www.ebay.com/n/all-categories

as input the complete set of child concepts (and in the case of Gen-Taxo++ also the parent concept) for the masked concept; in effect we have already identified the intermediate concept in the input.

Secondly, we predict the parents and children of the candidate concept labels generated in the first task. This corresponds to RQ3. For GenTaxo++, this task is performed by the conventional taxonomy completion model.

Following the spirit of RQ2 and RQ3, we only compute the relevant metrics on label quality and subsumption prediction with the subset of masked nodes that each method has successfully identified. This subset could be different for different methods.

*5.1.3 Metrics.* For RQ1, we follow the metrics used in GenTaxo, i.e., **Precision** (P), **Recall** (R) and **F1**. Precision is defined as the number of correctly recovered nodes divided by the total number of generated nodes, and recall is defined as the number of correctly recovered nodes divided by the number of masked nodes.

For RQ2, we adopt **BERTscore** [40], a representation-based metric that captures semantic correlations, instead of the exact match accuracy used in GenTaxo. This is because exact match with the reference (ground truth) is not necessary for regarding a generated label as good. For instance, a prediction of "footwear" where the reference is "shoes" would satisfy most practical demands and should be accepted. Such a prediction would still achieve a high BERTscore, but zero exact match accuracy. The model used to evaluate AliOpenKG results is `bert-base-chinese`, and the model used for eBay results is `roberta-large`. The metric contains three scores: **Precision** (denoted Bs-P), **Recall** (Bs-R), and **F1** (Bs-F1). Notice that while BERTscore claims contextual understanding and generally aligns better with human natural language understanding, it is computationally more intensive and lacks interpretability. However, it is the most appropriate metric we find for this evaluation given the lexical variations in labels.

For RQ3, we measure **Precision** and **Recall** of the predicted edges. Recall is defined as the proportion of the masked node's direct parent and children that are successfully recovered:

$$R = \frac{|(p(q) \cup c(q)) \cap (A_{pred}(q) \cup D_{pred}(q))|}{|(p(q) \cup c(q))|} \quad (2)$$

Precision is defined as the proportion of predicted edges that are factually true. However, our previous argument has established that the taxonomy itself is insufficient at expressing all true subsumptions. Therefore, we verify all edge predictions with qualified human judge contributors. Since the nodes in both taxonomies are e-commerce categories, we give the contributors auxiliary information on these categories (e.g., browse pages[6] for items in that category) to assist their judgment. Details of the human evaluation will be presented in Appendix D. With precision and recall, we also calculate and report **F1**. We do not report ranking based metrics which are commonly used in other taxonomy completion works, such as Mean Reciprocal Rank (MRR) and Hit@$k$, because ICON and ChatGPT are not ranking-based.

## 5.2 Main Results

We now present our evaluation outcomes in Table 3. As previously mentioned, the three GenTaxo-based baselines only differ in the

[6]For instance, https://www.ebay.com/b/1059 is the browse page for Men's Clothing

edge prediction task, and the two previous tasks are completed with GenTaxo alone.

For RQ1, we observe that despite having less information on the masked implicit concepts, ICON claims a solid 14% advantage over ChatGPT. This is partially explained by the fact that ChatGPT does not have access to the entirety of taxonomy structure (see Appendix C). Because of the way GenTaxo is used, it essentially faces a binary classification problem for each candidate position (either accept or reject the candidate position). GenTaxo's recall on this problem depends solely on one of its hyperparameters, negative sampling rate $r_{neg}$. In fact its recall approaches 1 when $r_{neg} = 0$. However, we choose $r_{neg}$ differently so that it optimises RQ2 metrics, as maximising recall for this binary classification problem is trivial. Precision for GenTaxo is always 1 because GenTaxo always generates one node for each accepted candidate position, and we do not have a sound mechanism to *reject* nodes: verifying whether a generated node is indeed an implicit concept is a difficult problem, and any human labelling attempt at it would be impeded by ambiguity and the vast amount of undecidable cases.

For RQ2, both ChatGPT and ICON are capable of generating very high quality labels. ICON outperforms ChatGPT slightly due to more comprehensive fine-tuning. ChatGPT uses verbal prompts with examples, which is less effective at adapting its style of generated labels to the style of reference. The GRU model used by GenTaxo lags in the competition, and in particular scores lower precision compared to recall. This is because the GRU in our experiment tends to generate excessively long labels.

For RQ3, our enhanced traversal-based search achieves the highest metrics while using the most lightweight model (other than TaxoExpan), thus displaying the strength of replacing a single prediction (as in previous works [12, 35]) with multiple graph-guided predictions. ChatGPT achieves similarly high precision due to its grasp of common sense knowledge which makes illogical subsumption predictions unlikely. However, ChatGPT suffers from the problem of searching through a large corpus, lowering its recall.

Performance is roughly similar across the two datasets, with ICON and GenTaxo++'s performance being slightly better on eBay due to its larger size and English labels: its larger size gives the models more fine-tuning data while its English text allows us to use PLMs pre-trained on larger corpora. ChatGPT on the other hand, is barely affected by either difference.

## 5.3 Ablation Studies

In order to attain high flexibility, ICON has many hyperparameters or settings during every phase of its workflow, some of which we leave to Appendix A due to their relatively minor influence. Here we study the effects of the following hyperparameters, which we find to be the most significant: search tolerance, forceful inclusion, and search space restriction.

The effects of varying each hyperparameter are shown in Figure 5. We conduct these experiments on the eBay dataset, with all other hyperparameters specified in Appendix B.

*Tolerance.* As tolerance increases, the algorithm traverses through a larger portion of the search space, reaching a nearly exhaustive search at $\tau = 4$. Due to imperfections of the SUB model, increased tolerance will monotonically enlarge the sets of predicted parents

**Table 3: Main evaluation results. Boldface marks the best performers in each metric.**

| Method | | implicit concept identification | | | concept name generation | | | concept insertion | | |
|---|---|---|---|---|---|---|---|---|---|---|
| | | *AliOpenKG* | | | | | | | | |
| | | P | R | F1 | Bs-P | Bs-R | Bs-F1 | P | R | F1 |
| GenTaxo++ | TaxoExpan | | | | | | | 0.356 | 0.347 | 0.352 |
| | TMN | **1.000** | 0.680 | **0.809** | 0.711 | 0.822 | 0.763 | 0.516 | 0.553 | 0.534 |
| | QEN | | | | | | | 0.557 | 0.601 | 0.578 |
| ChatGPT | | 0.401 | 0.730 | 0.519 | 0.924 | 0.915 | 0.919 | 0.762 | 0.625 | 0.686 |
| ICON | | 0.544 | **0.859** | 0.665 | **0.938** | **0.942** | **0.940** | **0.805** | **0.737** | **0.769** |

| Method | | implicit concept identification | | | concept name generation | | | concept insertion | | |
|---|---|---|---|---|---|---|---|---|---|---|
| | | *eBay* | | | | | | | | |
| | | P | R | F1 | Bs-P | Bs-R | Bs-F1 | P | R | F1 |
| GenTaxo++ | TaxoExpan | | | | | | | 0.378 | 0.368 | 0.373 |
| | TMN | **1.000** | 0.668 | **0.801** | 0.749 | 0.873 | 0.806 | 0.530 | 0.571 | 0.549 |
| | QEN | | | | | | | 0.562 | 0.609 | 0.584 |
| ChatGPT | | 0.406 | 0.734 | 0.523 | 0.927 | 0.921 | 0.924 | 0.777 | 0.616 | 0.687 |
| ICON | | 0.563 | **0.887** | 0.689 | **0.946** | **0.955** | **0.950** | **0.831** | **0.751** | **0.789** |

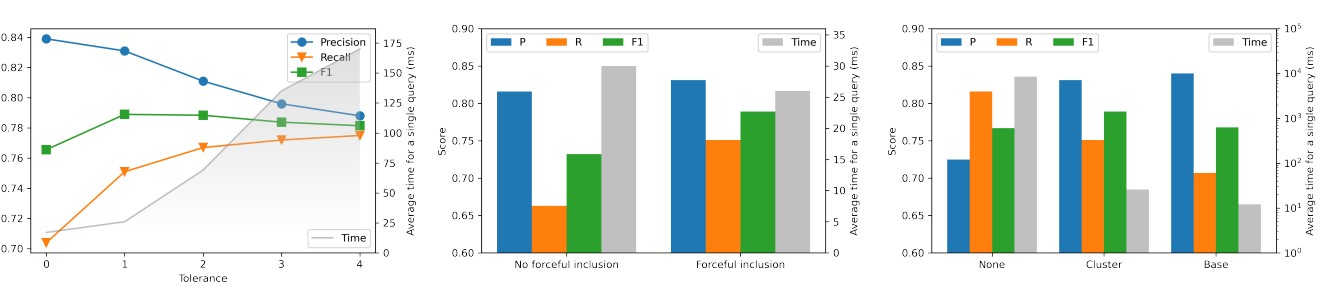

(a) Tolerance                               (b) Forceful inclusion                               (c) Space constraint (time in log scale)

**Figure 5: Effects of varying key search parameters based on the eBay dataset**

and children, shifting towards recall in the P-R tradeoff. The optimal F1 is achieved near $\tau = 1$. The average time spent on each candidate is displayed by the gray curve.

*Forceful inclusion.* Enabling this option will reduce the time cost slightly since the base concepts and their LCAs no longer require model inference. We notice in Figure 5b that both precision and recall are improved by forceful inclusion, since without it the SUB model has a chance to reject these correct subsumptions. However, this is partially an artefact of the evaluation setup. In real applications, there is no guarantee that the candidate, only represented by its label which is generated by a language model, faithfully represents its expected semantics of being a union of the base concepts.

*Search space restriction.* Restricting search space in the taxonomy brings tremendous speed improvement: cluster-level restriction makes the search about 300x faster, and base-level restriction is over 700x faster. The average search space is 20,322 concepts without restriction, 38 concepts with cluster-level restriction and 10 concepts with base-level restriction. Eliminating 99.5% of the nodes from the search space proves to not only gain speed massively but also improve the overall F1, demonstrating the strength of our restriction mechanism.

## 6 CONCLUSION

In this paper, we study the phenomenon of implicit concepts within taxonomies, and propose the task of implicit taxonomy completion. Three sub-tasks are required to perform implicit taxonomy completion. By mapping each sub-task to a well studied research task, we develop ICON, a system to automatically find and insert implicit concepts via enumerating intermediate concepts and enhanced traversal. ICON organically integrates three component models and features high malleability, i.e., it allows flexible control over its search behaviour and supports various levels of human involvement. Extensive evaluation has verified the strength of ICON when applied to real world taxonomies.

In future work we plan to extend this work to a larger domain, in particular general Knowledge Graphs (KGs) and ontologies. The core idea of enhanced traversal, which is using hierarchies to prune search branches, could be applied to any semantic relation that induces a hierarchy. Searching on multi-relational KGs would, however, require a different approach to constructing intermediate concepts and new ways of restricting the search space, suggesting promising directions for future research.

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

## A  PARAMETERS AND SETTINGS OF ICON

ICON allows for a variety of settings and hyperparmaeters to adjust its behaviour. We provide the complete list as follows:

- Overall control:
  - mode: Allows three options, "auto", "semiauto", and "manual", as described in Section 4.1.3. Defaults to "auto".
  - max_cycle: Maximal number of outer loops permitted, used in auto mode to ensure termination. Defaults to None which does not limit the number of outer loops.
- Implicit concept identification:
  - retrieve_size: The maximal number of neighbours ($k$) for the KNN model. Defaults to 10.
  - subset_size: The maximal size of subset ($m$) for the enumerated intermediate concepts during step 3 in Figure 3. Defaults to 2.
  - restrict_subset: Whether to limit the enumerated intermediate concepts to only those including the seed concept. Defaults to False.
- Concept name generation:
  - filter: Whether to ignore the intermediate concepts that have *trivial LCAs*: An intermediate concept $I = \{n_1, \ldots, n_s\}$ is said to have trivial LCA if $LCA(n_1, \ldots, n_s) \in I$. Such an intermediate concept is often of reduced value because it tends to coincide with its LCA concept. Defaults to True.
  - ignore_labels: The set of GEN model outputs which the system considers as rejection of the intermediate concept. Defaults to ["", "root", "NULL"].
- Search domain restriction:
  - subgraph_span: Allows three options, "All", "Base", and "Cluster". The search domain will be limited to the subgraph induced by the nodes that transitively subsume at least one concept in subgraph_span. Defaults to Base.
  - subgraph_crop: Whether to further restrict the search domain to the LCA of subgraph_span and the LCA's descendants. Defaults to True.
  - subgraph_force: If enabled, the search domain will always include the LCA of base classes w.r.t. the original taxonomy even after the taxonomy has been modified during previous iterations of ICON. Furthermore, descendants of the original LCA that are in the subgraph specified in subgraph_span will also be included. Does not take effect if subgraph_crop is False. This option ensures that the search domain does not exclude useful concepts when previous iterations of ICON creates new LCA concepts that are more specific than the original LCA. Defaults to True.
- Enhanced traversal:
  - tolerance: See Section 4.1.2. Defaults to 0.
  - force_known_subsumptions: See Section 4.1.2. Defaults to True.
  - force_prune_branches: Whether to force the search to mark all the descendants of a tested non-parent as visited in the top-down search, and mark all the ancestors of a tested non-child as visited in the bottom-up search. Since marking takes time, this will slow down the search if the search space is roughly tree-like, but potentially speed up the search if

more time is saved by not running the SUB model on the marked nodes. Defaults to False.

## B  IMPLEMENTATION DETAILS

In all experiments, hyperparameters of ICON are set to the following unless otherwise specified: cluster size $k = 10$, subset size $m = 2$ with no obligation to include the seed, tolerance $\tau = 1$, forceful inclusion is enabled, search space is restricted to the spanned by the cluster. The AdamW optimizer [17] and the linear learning rate scheduler with 400 warmup steps are used to train all models. During the contrastive learning of KNN model, minibatch size is 8, learning rate is 5e-5 and temperature $\tau = 0.05$. For both the training of T5 and BERTSubs, learning rate is 5e-5 and batch size is 16.

For GenTaxo++, we apply the authors' recommended settings for GenTaxo, TaxoExpan, TMN, and QEN. The only exception is Gen-Taxo's negative sampling rate, which is set to be 0.3 on AliOpenKG and 0.35 on eBay.

The ChatGPT we use is version 3.5 served during April 2023 and May 2023, and accessed via the OpenAI API. The running time statistics in Section 5.3 are obtained on our hardware with 6-core Intel Core i9 CPU and 1x V100 GPU.

## C  CHATGPT PROMPTS

We use two prompt templates, where the first template corresponds to implicit concept identification and naming (RQ1 and RQ2), and the second corresponds to concept insertion (RQ3). Each template consists of a header and a list of tasks. In the header, we give detailed instructions on the task and offer examples , and inform ChatGPT on the input style of the task list. The task inputs are minimal textual descriptions of each individual task. We do not repeat the task objective in the task inputs in order to reduce the amount of transmitted tokens and therefore the cost.

For implicit concept identification and naming, we illustrate our prompts with the following dialogue:

- **User**: You will be creating intermediate concepts from a list of concepts. An intermediate concept is a concept that accurately summarises a group of concepts, and has realistic meaning. For example, given the following list of concepts:
  1. Men's Clothing
  2. Men's Shoes
  3. Women's Clothing
  4. Women's Shoes
  We can create the following intermediate concepts:
  1. Clothing
  2. Shoes
  The first comes from Men's Clothing and Women's Clothing, while the second comes from Men's Shoes and Women's Shoes. However, not every list of concepts allows for a meaning intermediate concept. For instance, the following list:
  1. Duck Collectibles
  2. Smartphone Cases
  3. Industrial Coffee Machines
  4. Retail Display Cases
  Doesn't give any useful intermediate concept. Even though smartphone cases and retail display cases both contain the word "cases", they are very different types of entities

and shouldn't be grouped together. In those cases you are
supposes to answer "None".
I will be giving you a list of concepts and you will
answer with either a list of one or more intermediate
concepts, or "None". Answer "ready" to proceed.
- **ChatGPT**: I'm ready to proceed. Please provide the list
of concepts, and I'll generate the intermediate concepts
or indicate "None" as appropriate.
- **User**:
    1. Fishing Bait Buckets
    2. Snowshoes
    3. Fishing Nets
    4. Pottery & Glass Bowls
- **ChatGPT**: None
- **User**:
    1. Kids' & Teens' Bedroom Furniture Sets
    2. DVDs & Blu-ray Discs
    3. Antique Bedroom Sets
    4. Men's Vintage Clothing
- **ChatGPT**:
    1. Bedroom Furniture Sets
    2. Media Discs
    3. Vintage Furniture
    4. Vintage Clothing

Note that ChatGPT can output answers according to the format specified in the prompt, while the identified intermediate concepts are not perfect. For example, "Fishing Bait Buckets" and "Fishing Nets" could be summarised into "Fishing Equipments"; "Media Discs" does not summarise two or more of the given concepts.

For concept insertion, we illustrate our prompts with the following dialogue:

- **User**: You will be finding the parent concepts and child
concepts for a given concept in the taxonomy. I will
first give you the taxonomy by listing its concepts and
parent-child relations, and then give you the candidate
concepts to be worked with. For example, suppose we
have the following taxonomy, where the text outside the
bracket indicates a concept's name and the text inside the
bracket indicates the concept's position in the taxonomy:
    1. Everything (root concept)
    2. Clothing, Shoes & Accessories (child of 1)
    3. Sporting Goods (child of 1)
    4. Men's Clothing, Shoes & Accessories (child of 2)
    5. Women's Clothing, Shoes & Accessories (child of 2)
    6. Men's Clothing (child of 4)
    7. Men's Shoes (child of 4)
    8. Women's Clothing (child of 5)
    9. Women's Shoes (child of 5)
    10. Men's Athletic Shoes (child of 7)
    11. Women's Athletic Shoes (child of 9)
And we have the candidate concept "Sporting Shoes", the
correct answer would be:
child of 2, 3
parent of 10, 11
Since Sporting Shoes is an instance of both Clothing,
Shoes & Accessories and Sporting Goods, while Men's
Athletic Shoes and Women's Athletic Shoes are both instances
of Sporting Shoes. Always make your choice of parents as
specific as possible and children as general as possible.
Now I will give you the taxonomy, and since it's too long
for a single input I will split it by 50 concepts per

input, until I indicate the end with "this is the end of
the taxonomy". All you have to do is answer "keep going".
Then I will give you one concept per input and you are
going to answer in the format that I have shown above.
Answer "ready" to proceed.
- **ChatGPT**: Sure, I understand the task. Please provide the
first 50 concepts in the taxonomy to get started.
- **User**:
    1. Everything (root concept)
    2. Antiques (child of 1)
    3. Art (child of 1)
    (**Proceeds to list 50 concepts**)
- **ChatGPT**: keep going
- **User**:
    (**User proceeds to list all the concepts**)
    (...)
    20321. Scooters & Mopeds (child of 20312)
    20322. Snowmobiles (child of 20312)
    20323. UTVs (child of 20312)
    this is the end of the taxonomy
- **ChatGPT**: keep going
- **User**: Shirt
- **ChatGPT**:
    child of 12
    parent of 2646, 2795

Note that here concepts 12, 2646, 2795 refer to "Clothing, Shoes & Accessories", "Men's Shirts", and "Women's Shirts", respectively.

## D HUMAN LABELLING

We conduct human labelling online via the platform Appen[7]. The contributors are qualified workers registered at the platform. Each contributor is shown with task examples and quizzed before his / her judgments could be trusted and used. The quiz consists of 20 hand-crafted concept pairs that are highly representative of subsumption prediction problems, ranging from easy to hard. 2,002 contributors attempted at the task, but only 1,889 passed the quiz, and 1,871 produced trusted work. We collected 52,314 human judgments on 17,518 subsumption predictions, of which 17,145 predictions obtained three trusted judgments and are used in our estimation of the precision.

The information given to contributors in quizzes and actual tasks adopts the following template:
Category 1: <concept 1 label> <concept 1 browse page>
Category 2: <concept 2 label> <concept 2 browse page>
For example:
Category 1: Men's Vintage T-Shirts www.ebay.com/b/175781
Category 2: Men's T-Shirts www.ebay.com/b/15687
Where browse pages are webpages that display the items of an e-commerce category (an instance of what we call concept in this paper) and the category's related concepts in the taxonomy. For concepts that are generated by models and do not correspond to an e-commerce category, we only provide their labels. Contributors are instructed to read the labels and check the browse pages whenever applicable, and decide whether the first concept is a child of the second concept. In particular, the expected answer is negative when one or both of the labels under consideration do not make sense as e-commerce concepts.

---

[7]https://appen.com/

