# OpenReview forum: "Taxonomy Completion via Implicit Concept Insertion"
_ACM.org/TheWebConf/2024/Conference — TheWebConf24 Oral_

### Official Review · Reviewer_C743 · 2023-11-16

**Novelty:** 5
**Technical Quality:** 5

**Review:**

This paper proposed the algorithm to discover implicit concepts in taxonomies with deep learning methods.
The unique feature of the work is the combination of three different learning models corresponding to the step of clustering, generation of  new concepts, and locating them in taxonomies. The result show positive comparing with the existing methods and Chat-GPT.
The paper is well written technically so that the technical achievement is clear.
On the other land, the limitation of applicability is not clear. One reason is It that the targeted datasets are only e-commerce ones so that it is unclear whether the proposed method is meaningful beyond e-commerce domain. The other is that the targeted concepts are only the second layer concepts from the bottom in taxonomies. It is not certain that it would be applicable to other concepts.
The more fundamental question is whether it is really discovery of implicit concepts. The experiment is conducted to evaluate how it can re-discover artificially hidden concepts in the taxonomies. It is meaningful as the performance of the algorithm but not as the discovery of new hidden concepts.
Rather, RQ1, RQ2, and RQ3 with the experiment indicate how it can refine human-curated taxonomies automatically. The evaluation for RQ2 allows semantic similarity in concept names and one for RQ3 includes the discovery of implicit sub-concepts. The refinement of taxonomies itself is the important task so that the work would be re-organized for this direction.

**Questions:**

1.
Do you have any indication how the algorithm would discover new hidden concepts rather than the re-discovery of hidden concepts?
2.
Have you tested it with datasets in other domains, in particular, those with deeper hierarchy? The datasets in the paper are shallow as taxonomy.

**Reviewer Confidence:**

4: The reviewer is certain that the evaluation is correct and very familiar with the relevant literature

**Scope:**

3: The work is somewhat relevant to the Web and to the track, and is of narrow interest to a sub-community

---

### Official Review · Reviewer_og6N · 2023-11-22

**Novelty:** 4
**Technical Quality:** 5

**Review:**

The paper proposes a solution for a problem that the authors define as “implicit concept insertion” which consists on adding concepts to an incomplete taxonomy. The authors apply their approach to 2 real world taxonomies: one from eBay in English and another from alibaba in Chinese.


The authors divide the task in 3 steps: identifying implicit concepts, naming implicit concepts and inserting them in the taxonomy. Those steps are further detailed and the authors describe the different algorithms and approaches employed.


I appreciate that the paper is very well written and I didn’t find any important typo. The motivating example with clothes is clear and the description of related work and preliminaries is also useful. Maybe one minor point is that section 3.2 titled “problem statement” only refers to the insertion of a concept q in the taxonomy and not the whole problem statement described in the paper.

I found quite interesting that the authors also compare their results with ChatGPT although looking to the prompts described in appendix C, I am not quite sure about the quality of the results.

One issue for me is that the authors don’t include a reference to any source code or dataset that could be used to check the reproducibility of their approach. The authors include a simple section "implementation details" in appendix B and another one about the hyperparameters in appendix A but there is not further details about the implementation and its availability.

After reading the paper, I wonder if the results could be improved with the use of multilingual taxonomies, i. E. taxonomies whose concepts could be labeled, for example, in both Chinese  and English.

**Questions:**

- Are the source code/datasets available?

**Ethics Review Description:**

I think there are no ethical issues, although the authors indicate that they used the platform appen to obtain human labelling. I am not sure if that could have some ethical concern.

**Reviewer Confidence:**

1: The reviewer's evaluation is an educated guess

**Scope:**

3: The work is somewhat relevant to the Web and to the track, and is of narrow interest to a sub-community

---

### Official Review · Reviewer_c1vu · 2023-11-23

**Novelty:** 5
**Technical Quality:** 6

**Review:**

The authors study the phenomenon of implicit concepts within taxonomies and propose the task of implicit taxonomy completion along three sub-tasks in a semiautomatic manner supported by a system called ICON which combines entity retrieval, text summary, and subsumption prediction approaches. The authors tested and evaluated their framework against real world taxonomies with promising results.

Quality:
The paper addresses a very interesting, obviously under researched topic at a high formal level. The research questions are well defined, formalized and operationalized, and an evaluation of the proposed framework has been carried out. The research is well contextualized in the related work and white spots in the state of the art, limitations of the proposed solution and future work have been addressed. Nevertheless, there is room for improvement in the latter aspect, preferably with a more substantial reference to the convergence of graph-technologies and LLMs for the specific purpose of this paper.

Clarity:
The paper is well written, and all addressed concepts are well described and formalized, especially given the supplementary material provided in the Appendices of the paper. It is a pity that the framework has not been released under an appropriate license for testing purposes (at least not information on this is available in the paper).

Originality:
The paper introduces a sound approach for implicit concept detection and taxonomy enrichment which is worth discussing. Although the claim that there is little work available in this area can be confirmed the authors should additionally scan the following literature for relevance:
https://arxiv.org/abs/2202.00070
https://www.iti.gr/~bmezaris/publications/csvt19_preprint.pdf
Gözüaçik, Ö., & Can, F. (2020). Concept learning using one-class classifiers for implicit drift detection in evolving data streams. Artificial Intelligence Review, 54, 3725 - 3747.

Significance:
The paper contains relevant questions and methodological inputs contributing and advancing the subject matter.

**Questions:**

Given the growing interest in the convergence of graph-based data representation and probabilistic methods, can you be more specific on how graph-based approaches shall and could be used to improve your suggested approach? Where do you see benefits? What are the trade-offs?

**Ethics Review Description:**

There are noo ethical issues related to this topic.

**Reviewer Confidence:**

2: The reviewer is willing to defend the evaluation, but it is likely that the reviewer did not understand parts of the paper

**Scope:**

3: The work is somewhat relevant to the Web and to the track, and is of narrow interest to a sub-community

---

### Official Review · Reviewer_de3b · 2023-11-23

**Novelty:** 6
**Technical Quality:** 4

**Review:**

The paper aims to complete  existing taxonomies by 1) discovering implicit concepts that can be hidden, 2) naming them and 3) inserting them at the right place. The idea of “implicit concept” is new compared to the state of art. The proposal named ICON for  Implicit CONcept Insertion, combines different well-known algorithms to solve the three tasks : Entity retrieval and KNN for discovering implicit concepts, text summarization for naming the implicit concepts, and BERTSubs to  insert new concepts in the right place.
The approach is validated on two real taxonomies: the one from ebay in english and the other one form AliOpenKG in Chinese.

Strong points:
* The paper is relevant for the track Semantic and knowledge of the conference.
* The paper is nicely written,  with convincing examples, and easy to follow.
* The idea of implicit concepts is original and interesting.
* The proposal is well positioned vs SOTA approaches.

Weak points:
* Thanks to the (nice) examples of the introduction, we have an intuitive understanding of what is an implicit concept, but there is no real definition of what it is. As it is, implicit concepts are defined by examples. It is written “careful identification of intermediate nodes is required in order to find useful implicit concepts that can improve the taxonomy’s quality and benefit downstream applications”. I fully agree with that, but what are the expected formal properties to be filled by an implicit concept?

* The proposal in section 4 is presented mainly by explaining *how* the three subtasks are realised, but not really why it should be done like that, especially for the first subtask : implicit concept discovery. For example, it is not obvious for me that implicit concepts should appear in KNN-cluster: It is written : “The reason we select similar concepts is to  increase the chance a subset built from these concepts identifies an  implicit concept.” Does this method  identify all “good” implicit concepts or just some ?? How can we be sure that we discovered an interesting implicit concept like that?

* The experiment validates the proposal by artificially creating implicit concepts by masking” some of the existing taxonomy concepts. I agree that this method is a way to validate the ICON approach, but it does not give evidence of implicit concepts that may not exist in Ebay and AliOpenKG. It is a serious issue for me as we cannot see the impact of ICON on existing taxonomies. I expect more than just validating the approach. I expect to see how ICON can improve existing taxonomies by inserting new pertinent concepts in the taxonomy. To be more general, if implicit concept is an appealing idea, we don’t know if its usage change marginally or drastically existing taxonomy ie. We don't know the usefulness of the whole approach.

**Questions:**

* How ICON impact the Ebay taxonomy? How many useful new implicit concepts you discovered in ebay? Can we see them somewhere?

**Reviewer Confidence:**

3: The reviewer is confident but not certain that the evaluation is correct

**Scope:**

4: The work is relevant to the Web and to the track, and is of broad interest to the community

---

### Official Review · Reviewer_GSGN · 2023-11-28

**Novelty:** 4
**Technical Quality:** 5

**Review:**

The paper describes a technique for integrating taxonomies with additional terms emerging from existing terms in the taxonomy. It has been evaluated using popular large taxonomies available in the world of online commerce, which helps place this work aptly in the context of TheWebConf and of this track.

The paper is overall well-structured and easy to follow, without significant language issues (only, please correct line 15 in the abstract: "based *on* the existing concept structure". This helped formulate a few points where further clarification would be helpful and help a re-assessment of this work which is otherwise valid and pertinent. See the following Questions:

**Questions:**

- The Related Work section is a little disappointing in that it doesn't quite offer a critique of the state of the art, highlighing where the proposed approach differs and is expected to be novel or overcome limitations (also GenTaxo is only discussed outside that section). I don't think it would be very burdensome or lengthy to adapt the section: would it be possible to do so?
- Where does the Tolerance factor described in page 4 fit in Algorithm 1? Is it part of the SUB_MODEL?
- Where an evaluation is carried out in terms of precision, recall and F-Score, it is not clear to me how exactly the necessary correctness is assessed. Was it a user evaluation? Does the Human Labelling section in Appendix D have anything to do with it? Please explain.
- I get the GenTaxo as a term of comparison, but why evaluate this approach against ChatGPT which, as admitted in the evaluation section, suffers from a handicap (size of corpus, not being re-trainable) that doesn't place it optimally in the use case at hand?

**Ethics Review Description:**

-

**Reviewer Confidence:**

3: The reviewer is confident but not certain that the evaluation is correct

**Scope:**

3: The work is somewhat relevant to the Web and to the track, and is of narrow interest to a sub-community

---

### Decision · Program_Chairs · 2024-01-22

**Decision:**

Accept (Oral)

**Comment:**

The paper proposes a method to extend an existing (incomplete) taxonomy by inserting new concepts in its hierarchical structure. The paper is clear, and proposals are innovative and quite appropriately tested on relevant taxonomies (even if the evaluation could be more convincing, or at least convincingly written). The topic is quite specific, but as many in the community do work with taxonomies that are very imperfect, the paper should be read with interest by more than a few.